# WeChatting for Health: What Motivates Older Adult Engagement with Health Information

**DOI:** 10.3390/healthcare9060751

**Published:** 2021-06-18

**Authors:** Xiaoxiao Zhang, Xiaoge Xu, Jiang Cheng

**Affiliations:** 1School of International Communication, Faculty of Humanities and Science, The University of Nottingham, Ningbo 315100, China; hnxxz6@nottingham.edu.cn; 2The Institute of Mobile Studies, The University of Nottingham, Ningbo 315100, China; Xiaoge.Xu@nottingham.edu.cn; 3School of Education, Peking University, Beijing 100871, China

**Keywords:** social media, health information, older adults, engagement behaviors, health communication

## Abstract

Although WeChat has become increasingly popular among Chinese elderly people as a tool to engage with health information, little research has examined their motivations for health purposes and their engagement with health information on the site. By applying the two-stage Use and gratification (U&G) approach, we first conducted in-depth interviews (*n* = 20) to explore older adults’ distinctive motives. Based on the 22 motives found in the qualitative research, we developed a questionnaire for an online survey (*n* = 690) to further investigate how these motives affect older adults’ engagement with health information on WeChat. As the result, six motive typologies were identified: information needs, social support, surveillance, social interaction, self-agency building, and technological convenience. Together, these six types of motivations jointly account for 59.9% of the variance in older adults’ engagement with health information (M = 2.71, SD = 0.79, adjusted *R*^2^ = 0.59, *p* < 0.001). Social support and information needs were significant predictors, suggesting that the older WeChat users’ active engagement is driven by personal instrumental gratification. This study examines the explanation power of U&G theory in a health context, as well as provides the practical implication for leveraging mobile social media to improve older people’s healthcare management.

## 1. Introduction

Owing to growing digital technologies and the increased adoption rate of mobile devices, mobile social media has become an important alternative tool to aid in peoples’ daily health management [1]. In the present, people can easily access various health and medical information from social media, especially when they are dissatisfied with a doctor’s healthcare suggestions and health information [2]. Social media platforms open up a new pathway for health communication and health promotion. With customized and timely online health information, people can better understand various health-related issues, which can result in better health decisions and better illness detection and prevention.

Coupled with the increasing adoption of smartphones, older people are increasingly turning to social media for diverse needs and gratifications [3]. Rather than being alienated by digitization, elderly Chinese people are embracing the possibilities of mobile digital technology [4]. Social media has become indispensable in their lives, and they engage with more digital content than ever before. In particular, older people use WeChat, a “must-have social media app”, which originally functioned, in essence, as an instant messaging tool. According to Tecent’s annual report [5], the number of active WeChat users aged 55 years old and above increased from 7.86 million to 50 million from 2016 to 2017.

WeChat appeals to almost 90% of Chinese smartphone users with its all-in-one approach [5], as it integrates a variety of functions such as sending messages, making video or voice calls, posting different content on Moments, reading news, ordering in a restaurant, shopping, and paying the bills, which does indeed make people’s everyday life easier. Today, WeChat has evolved beyond the above functions, and it is turning to a new frontier: healthcare. For example, WeChat official accounts provide users with customized and tailored health-related information and the information has been continuously re-transmitted to other users, providing WeChat with enormous potential to affect the public’s health status. Chinese older people begin to accept, WeChat, this novel way to manage their daily health. Their preferred health-related activities on WeChat include reading health education articles, seeking healthcare information and exchanging personal health experience with people have similar health conditions [6].

Although Chinese elderly people are increasingly turning to WeChat for health communication, little research has looked specifically at their motivations, and the subsequent behaviors on the site. It is highly necessary and worthy endeavor to examine their WeChat usage for health purpose as it can providing insights into leveraging mobile social media for health education and health promotion among the older population. Thus, the current study investigated older people’ distinctive desires of using WeChat for health purposes and how their motives facilitate health communication through their engagement behaviors on WeChat. By applying the two-stage procedure of uses and gratifications approach, we first conducted in-depth interviews to explore older people’s motives for health purpose. Based on the motives found from the interviews, we then developed a questionnaire for an online survey to further investigated older people’ subsequent engagement behaviors of health information on WeChat.

## 2. Background Literature

### 2.1. WeChat and Health

WeChat, owned by the Chinese tech company Tencent, is an “all-in-one” application that enables users to send messages, make video or voice calls, and post diverse content (i.e., text, pictures, and videos) through Moment feeds or exclusively to a single WeChat friend or a WeChat group. Today, WeChat has evolved beyond these basic functions, allowing users to complete a wide array of daily tasks: looking for apartments, buying tickets, transferring money, paying utility bills, or even getting access to third-party services through embedded mini-programs on one platform simultaneously. As of 2017, WeChat possessed 1.04 billion active users in over 200 countries, with half of the users using WeChat for up to 90 min per day [7]. Now, it has turned to a new area: healthcare management. During the past 10 years, WeChat has become “the biggest online healthcare destination in China” [8], offering a wide array of online health services to facilitate users’ health management, such as health education, health products, online medical diagnosis, as well as health insurance. As of 2019, 38,000 healthcare organizations and companies had opened official public accounts, publishing 500,000 health-related articles in 2020 [9], helping users better manage their health and cope with disease in their daily life. WeChat offers online medical diagnosis and telehealth through mini-programs. Users can schedule an online appointment with doctors for free and identify the appropriate hospital and doctor with the help of a medical virtual assistant on a mini-program. WeChat is now working with public hospitals to simplify operations by moving the process of hospital navigation, checking medical reports, and medical payment all onto WeChat. Moreover, WeChat launched their own healthcare knowledge platform, Tencent Medical Knowledge bank, providing user-directed health and medical education. It also offers group-buying options through a mini-program, allowing users from groups to access volume discounts and buy healthcare products in bulk. Recently, commercial health insurance services were added to the WeChat digital tool belt, allowing users to purchase insurance with a low minimum commitment.

Among all the medical and health services provided by WeChat, reading the health-related articles released by the WeChat official account is the most common health activity of WeChat users [10]. A recent national survey found that over 98% of smartphone users seek health information on WeChat, and one third of them read health education articles regularly offered by public official accounts [9]. It seems that WeChat has successfully triggered health behaviors among the general public from disseminating healthcare information to assisting in self-health management. In addition, WeChat’s unique interactive functions, such as like, share and comment, allow users to seek and share health information and generate health-related content to express their own health experiences and opinions on different health issues. These interactive functions of WeChat constitute an “information-rich environment”, making WeChat a popular health platform [11].

### 2.2. Uses and Gratifications of Social Media

Uses and gratifications (U&G) is considered one of the most effective theories for identifying people’s motives behind media use. U&G has been employed to examine how and why people use media, rather than what media does to people [12,13]. It posits that the individual “is active and intentional enough to select media to meet their distinctive demands” [14] (p. 20). U&G researchers usually speak of “motivations” when describing why users consume certain media and what satisfactions they eventually receive from media, e.g., [15,16]. Originally, Katz, Blumler, and Gurevitch identified six motive typologies that people satisfied via mass media: cognitive, affective, personal integrative, social integrative, and tension release [14]. Then, McQuail distinguished four gratification categories: diversion, personal relationships, personal identity, and surveillance [17]. Later, McQuail proposed an updated version that included social integration, social involvement, entertainment, information, and personal identity [18].

However, the interactive nature of new media has led to a re-examination of motives found in the original U&G study. Social media scholars have done a great deal of research on how and why people use this new medium. For example, Muntinga, Moorman, and Smit further proposed “empowerment” and “social pressure” as two motives, uniquely related to social media usage [19], in addition to the four motives posited by McQuail [18]. Quan-Haase and Young examined the gratifications obtained from Facebook and instant messaging, and the results indicated that people tend to seek “enjoyment” and “knowledge” about social activities from Facebook, while instant messaging users tend to build and maintain harmonious social relations with others [20]. Regarding the motives for WeChat usage, Huang and Zhang argued that the perceived need for information, sociability, and entertainment were significant predictors for the adoption of WeChat among Chinese users [21]. In addition to the need for information and socialization, Gan and Li argued that media appeal is also one of the important factors affecting an individual’s use of WeChat [22]. Chang and Zhu’s study further explored the motivational factors during the pre-adoption and post-adoption, which included conformity, sociality, information, and entertainment [23]. Chen et al. studied the information-sharing behaviors on WeChat Moment, and found that familiarity and identifiability in an interpersonal relationship will positively and significantly affect WeChat users’ information sharing behaviors [24].

Prior studies have provided sufficient evidence that U&G is an appropriate approach for investigating social media [13,25] and, most importantly, it helps to understand how motives can be significant psychological antecedents to explain cognitive processes behind various communication behaviors [26]. Thus, this study draws from U&G theory to explain older users’ motivations for engaging in WeChat for health-related purposes.

### 2.3. Motivations for Using Social Media for Health-Related Purposes

Health information exchange on social media makes people feel empowered and informed, which enables them to take an active role in their daily health management. Prior studies identified that the psychological motives of social media usage differ according to the context [27]. For example, compared to the general usage of social media, Moorhead revealed a different set of health needs: interactivity, greater availability of tailored information, access to health information, peer support, public health surveillance, and the potential to influence health policy [1]. In Newman et al.’s research, seeking emotional support was also an important gratification sought through Facebook interactions regarding weight loss and diabetes management [28]. Given the enormous potential of WeChat for health communication, examining people’s motives for adopting WeChat for health-related purposes is both necessary and important.

Although some motives of social media usage, as suggested by previous research, might be applicable for health-related needs and purpose on social media, a distinctive typology of health-related motivations should be developed. Compared to other social media (i.e., Facebook and Twitter), WeChat users at least know their WeChat contacts to some extent in reality as all WeChat users need personal approval to access to others profiles and post. Such a private communication environment may develop distinctive motivations among users. In addition, U&G scholars continually revealed the age-related differences in social media usage, which provides important implication as to how to identify social media usage among specific populations [29,30]. For instance, Kathryn and Madde found that senior citizens use social network sites because they want to rebuild the connection with others in their age group as well as obtain social support from online friends [31]. Jung and Sundar investigated older users’ gratifications obtained from specific affordances on Facebook and they found status updating, personal stories sharing were positively associated with community-building gratifications, and conversations and comment play an important role in providing interaction gratifications [32]. Righi, Sayago and Blat also found that photo-sharing is a significant reason among the elderly [33]. Given that the motivations specifically related to older people should be identified in this study, it is logical to speculate a set of different motivations that may be engendered among them for health communication on WeChat. Therefore, to fill the research gap and understand the potential of WeChat for fulfilling older people’ health-related needs, we proposed the following research question:

RQ 1: what are the motivations that affect older peoples’ adoption of WeChat for health-related purpose?

### 2.4. Engagement Behaviors on Social Media

The U&G theory perspective posits that active audience select media to fulfil their distinctive desires and goals, which develops not only multiple motives, but also a different set of cognitive, affective, and behavioral outcomes [34]. That is, people’s intrinsic needs are manifested in their expressed motivations and thus affect their subsequent communication behaviors such as engagement with media content. Engagement has been conceptualized as “a user-initiated action” [35] (p. 8), which leads to a co-creation of value [36], and it comprises not only behavioral actions but also cognitive thoughts, and emotional feelings aspects [37]. Calder and Malthouse pointed out that engagement is a state of involvement and connectedness between the user and the engagement object such as the media content, and they believe that engagement behaviors are triggered by motivational forces such as building social relationships and upgrading one’s status [37]. For example, research has confirmed that the gratifications of social interaction encourage people to actively share a common interest with others, while information gratifications activate people to deliberate over the political messages [38]. Besides, scholars found that gratification for technology convenience, information exchange and social involvement are important motivational factors for people’s civic engagement on social media [39]. These studies revealed that motivations for people’s needs, and gratifications can be significant predictors of online behavioral engagement. Thus, we argue that people are motivated to use WeChat for health-related gratifications, and they actively engage with health information to gratify their desires and needs.

The most common behavior people perform on WeChat is engaging with the information by means of updating the WeChat Moments, making text/voice chat, and reading articles on WeChat official account [22]. In addition, people usually not only consume information themselves, but also share this with friends or family members who were also looking for the same information [10]. Through people’ engagement with health information on WeChat, such as sharing or/and liking, they can access more healthcare information, and their health literacy and health beliefs can be also reaffirmed by their active interactivity online. Therefore, this study takes a further step to examine how older people’ motivations are associated with their engagement behaviors with health information on WeChat.

Social media engagement is typically manifested symbolically through actions like “sharing” a post, “liking” a post, or “commenting” on a post [40,41]. Accordingly, this study employed the suggested dimensions to investigate older users’ engagement behaviors of health-related content on WeChat. By looking at different dimensions of engagement behaviors separately, it is possible to examine the motivations that trigger these behaviors separately, thus obtaining a better understanding of older people’ engagement behaviors of health information on WeChat. The above claim leads to the second research question:

RQ 2: which types of motivations will best predict older peoples’ engagement behaviors (i.e., sharing, liking, and commenting) regarding health information?

## 3. Materials and Methods

This study adopted the classic two-stage U&G approach, which combines both qualitative (study1) and quantitative research (study2) [42]. Study1 explored health-related motivations through in-depth interviews. Then, study 2 developed a questionnaire according to the motivations found from study 1. An online survey has been conducted. Additionally, the experience materials in the in-depth interviews were used again to thoroughly explain the association between the motivations and engagement behaviors of health information among older WeChat users.

In terms of the definition of “older adults”, China’s retirement policy, that is, men over 60 years old and women over 55 years old [43]. However, we selected sample group aged from 50 to 75 years old, due to the reason that most older adults who actively use social media for social and information purposes are young elderly over 50 years old with higher income and higher education level [44]. We believed that broadening the age threshold will help us collect real and useful data.

The research ethics subcommittee of the University of Nottingham has approved this study. We informed participants of the research purpose, research process and ethical issues during the recruitment stage. Additionally, participants were informed of the principles of voluntary participation, anonymity, expected benefits and different ethical considerations before the research officially started. Specifically, all the participants in study 1 and study 2 were voluntary and anonymous, and no personal identification information was obtained from the research except age, gender and some other insensitive demographic information. In-depth interviews and questionnaires were conducted with prior consent, and the materials were used anonymously.

### 3.1. Study 1: Identify Older Adults’ Motivations from In-Depth Interviews

In study 1, the in-depth interviews were conducted. We first sent an informed consent form to older people who met the study criteria via WeChat. Our criteria included older people aged 50 to 75 years who lived in an urban area; used WeChat for at least 30 min per day; and regularly use (or have used) WeChat for health-related activities. Those who wish to participate in the interview indicated their interest in the consent form. Then, we contacted them directly to arrange for the interview through WeChat videos, telephone calls, or face-to-face communication. A total of 20 participants (55% Males, 50–72 years old) were selected using convenience sampling and snowball sampling. Before the formal interview, participants were given an interview guide with major questions to help them familiar with the entire process. The interview lasted about 30–40 min for each person and was recorded by our research assistant. At the end, participants were asked to fill out a demographic questionnaire that included information about their age, gender, educational background, health status and daily WeChat usage.

As for the data analysis, study1 first identified 22 distinct motivations for using WeChat for health gratifications. Qualitative data were analyzed using NVIVO 12. All interview transcripts were aggregated to one text for content analysis through an open coding process. Themes were developed through an axial coding process [45]. Similar responses were combined into the same motivations. For example, the responses like “to obtain up to date health information” and “to help me to access the newest health news” were combined. The second coder coded 60% of the transcripts to check inter-coder reliability. Using Cohen’s kappa that corrects for chance agreement, the agreement level in this study was 85%, which is an almost perfect agreement [46].

Drawing upon the interviews with 20 older WeChat users, the most frequent motivation mentioned was “to acquire health information” (*n* = 20) followed by “to obtain emotional and informational support from others on WeChat when talking about some health issues and personal health conditions” (*n* = 17). Other common motivations included: “to build and consolidate relationships by sharing important health information” (*n* = 13), “to express a personal stance on health issues makes me feel satisfied” (*n* = 10), and “to filter health-related misinformation and offer high-quality information for WeChat friends and groups” (*n* = 8). All 22 motivations are listed in Table 1.

### 3.2. Study 2: Examine the Relationship between Motivations and Engagement Behaviors

Based on study 1′s findings, study 2 designed a questionnaire that includes all motives identified from study1. The motivations were converted to questionnaire items that ask respondents to rate them on a five-point Likert scale (1 = strongly disagree, 5 = strongly agree). Fifty older people were invited to participate in a pilot test before the formal questionnaire was issued. Then, we revised the questionnaire according to their feedback.

A cross-national data was collected by a highly reputable research firm from 23 December to 3 January 2021. Our sample criteria included older people aged 50 to 75 years who lived in an urban area and regularly use WeChat for health-related activities. Participants were first asked whether they were WeChat users (73% answered yes). If the the answered yes, they began to answer the questions on the questionnaire. Additionally, we used trap questions, repeated IP detection and time control to eliminate invalid cases in the process of administering the questionnaires. We collected the 690 response, 47.5% (*n* = 328) were female; the average age was 60.47 (ranging from 50 to 74 years old); 87.68% received a middle school or higher degree; 75.79% had a monthly personal income between CNY 3001-5000. Table 2 provided the detailed demographic information of the participants.

Adapted from the literature on engagement [47,48], older adults’ engagement behaviors were mainly measured in three dimensions rated on a five-point scale (1 = Never, 5 = Very Often). Sharing was measured by two items: How often do you share health information to your WeChat friends or groups and how often do you share health posts on WeChat moments? (M = 3.01, SD = 0.90, α = 0.76); The frequency of liking was measured by two items: How often do you like a health post on WeChat Moments and how often do you like a health article when you read it on WeChat? (M = 2.76, SD = 0.89, α = 0.80); Commenting was also measured by two items: How often do you comment on or reply to a health post on WeChat Moments (M = 2.1, SD = 0.80, α = 0.77).

Motivations for health purposes were identified in study 1. Participants were asked to rate the 22 motivation statements on a 5-point Likert scales (1 = Strongly Disagree, 5 = Strongly Agree). Table 1 and Table 3 provide the full list of motivations.

WeChat usage scale was adapted from Chen’s work [49], including the following items rated on a five-point Likert scale (1 = Strongly Disagree, 5 = Strongly Agree): Using WeChat has become a part of everyday life, I always open WeChat to check Moments, message and public official account; I feel out of touch when I haven’t logged on to my WeChat for a while; I think I am a part of the WeChat community, which makes me feel a sense of belonging; I would be upset, If WeChat is shut down for a long time for some reason. Cronbach’s alpha was 0.81 (M = 3.76, SD = 0.77). As to the frequency of WeChat use, previous research found that more than 78% of older adults use WeChat around 1.37 h per day [5]. Accordingly, participants were asked to indicate their frequency of using WeChat (M = 3.04, SD = 0.58) on a five-point scale with specific time range: 1 = rarely use (less than 30 min), 2 = sometimes (30–60 min); 3 = often (61–90 min); 4 = frequently use (91–150 min); 5 = always (more than 150 min).

This study included some demographic variables (i.e., gender, age, education level, monthly income, and health status) as control variables. As previous research has indicated that some demographic variables are closely related to social media user’s engagement behaviors [10], so we control them to isolate the unique association between the investigated variables.

## 4. Results

To answering RQ1, 22 items were subjected to an exploratory factor analysis with principal component extraction and varimax rotation to test the factor structure of the motivation items. To further answering RQ2, several hierarchical regression analyses were conducted to examine the association between older adult’s motivations and engagement behavior of health information on WeChat.

### 4.1. Exploratory Factor Analysis of the Motivations for Health-Purposes

Study 2 was first conducted by an exploratory factor analysis on all 22 items in order to answer RQ 1. The KMO measure was found to be 0.892, and Bartlett’s test of sphericity was found to be significant. As a result, we labelled the six motivation factors based on the results extracted from 22 items, which together accounted for 72.66% of the variance. These factors are: health information needs, surveillance needs, social support, social involvement, self-agency building, and technological convenience. Two items were removed as they failed to meet the suggested criterion of including items that load at 0.50 or greater [50]. Table 3 provides the relevant information.

### 4.2. Hierarchy Regressions with Six Motivations and Engagement Behaviors on WeChat

To answer RQ 2, we conducted separate hierarchical regressions to explore what and how older peoples’ motives predict their engagement behaviors with health information on WeChat. Each regression analysis included the six motivation factors as the independent variables and the demographics and WeChat usage as control variables. Three dimensions of engagement behaviors served as dependent variables. Table 4 shows that five motivations, namely, surveillance needs, information needs, social involvement, social support, and self-agency building, were positively and significantly associated with older adults’ active engagement, jointly accounting for 59.9% of the variance in older peoples’ engagement behaviors (M = 2.71, SD = 0.79, adjusted *R*^2^ = 0.59, *p* < 0.001). The motivation of technological convenience did not show any statistical significance.

Specifically, three motivations were significantly and positively associated with sharing behaviors (M = 3.00, SD = 0.90), accounting for 52.8% of the variance (adjusted *R*^2^ = 0.52, *p* < 0.001). Health information needs (*β* = 0.25, *p* < 0.001) were the strongest predictor, followed by social support (*β* = 0.19, *p* < 0.001), and social interaction (*β* = 0.17., *p* < 0.001). For the liking dimension (M = 2.76, SD = 0.89), five motivations were found to be significantly and positively associated with older adults’ frequency of liking a health article or a health post, jointly accounting for 47.9% of the variance (adjusted *R*^2^ = 0.47, *p* < 0.001). Health information needs were also the strongest predictor of older peoples’ frequency of liking a health post or health articles (*β* = 0.24, *p* < 0.001). This was followed by social involvement (*β* = 0.18, *p* < 0.001) and social support (*β* = 0.15, *p* < 0.001). Self-agency building (*β* = 0.097, *p* < 0.05) and surveillance (*β* = 0.12, *p* < 0.001) were also significant predictors. For the commenting dimension (M = 2.39, SD = 0.91), motivation of information need (*β* = 0.18, *p* < 0.001) and social support (*β* = 0.15, *p* < 0.001) were the two most important factors, followed by social interaction (*β* = 0.15, *p* < 0.01). These three motivations accounted for 33.4% of the variance (adjusted *R*^2^ = 0.33, *p* < 0.001). Surveillance needs and self-agency building were found to have no obvious influence on commenting behaviors.

Additionally, we found that age and health status were negatively associated with the active engagement of health-related content on WeChat (*β* = −0.09, *p* < 0.05; *β* = −0.26, *p* < 0.001). Education level and WeChat usage were found to be positively associated with active engagement behavior on WeChat (*β* = 0.30, *p* < 0.001; *β* = 0.19, *p* < 0.001).

## 5. Discussion

In light of the classic two-stage U&G approach, this study identified older WeChat users’ motivations for health-related purposes as well as provide first hand empirical evidence on how their motivations predict the different dimensions of engagement behaviors with health information on WeChat.

Firstly, our results revealed a unique pattern of motivations for health-related purposes among older WeChat users. Six types of motivations, namely, information needs, social support, surveillance, social interaction, self-agency building, and technological convenience, were identified by rigorous statistical analyses across large group samples. More specifically, the motivation for seeking health information was positively associated with each dimension of older users’ engagement behaviors. This finding is consistent with previous studies [51,52,53]. Older WeChat users who are motivated by information needs are likely to perform engagement behaviors in terms of sharing, liking and commenting. This suggests that the more elderly people seek healthcare information, the more likely they are to participate in healthcare information dissemination and elaboration. Older people’ engagement behaviors with health information benefit their well-being and health management as WeChat provides a valuable opportunity for exchanging health experience and obtaining health education such as information on prevention, diagnosis, and treatment of specific conditions and disorder [54].

Moreover, older people use WeChat not only to obtain health information, but also for purposes of socializing [55,56,57], such as receiving healthcare suggestions from someone with similar health conditions and sharing useful health tips with families and friends on WeChat. This finding also echoes prior studies indicating that social media platforms are identified as places where elderly people can get together and make social contacts to overcome loneliness, get the latest information about family members and acquaintances, and establish or consolidate their social relationships e.g., [58,59]. For older Chinese, the one-child policy and the new urban migration policy have both led to them living far away from their children, causing a more stratified social structure. After retirement, they face anxiety and loneliness because of a lack of social and family support. As a result, despite WeChat is a fairly new technology for the elderly in China, it has been adopted to “build new patterns of social bonding” by many Chinese elderly people [60] (p.10), it offers older people the possibility to engage in meaningful social contact and provides more opportunities to give and receive emotional and informational support through their friend networks.

Notably, this study reveals that the motivation for surveillance is another indispensable component among the elderly. Such a motivation is distinctively found in our study, given that it has been little studied in previous research in the health communication field. We argue that older people seek surveillance needs through WeChat mainly because the unique information environment of WeChat as a platform for providing timely and comprehensive news regarding accidents, diseases, and disasters. From our in-depth interviews, we found that older people are especially concerned about large public events that are unusual and lethal, such as the corona-virus pandemic, given that these threats significantly affect their quality of life and personal well-being. WeChat provides up to date news about public health events, which satisfy the older people’ survival instinct to detect surrounding threats and allow a quick reaction to deal with a potential or real threat [61].

Moreover, we also found that the motivation of establishing self-agency is a unique and fundamental motives among older people. This particular motive has exerted a powerful relationship with older people’ s liking and commenting behaviors. Older people pay more attention to healthcare information, and they are considered to have more say in this field. Thus, the more an elderly people want to build their self-agency in health field, the more likely they engage in liking and commenting behaviors for social endorsement and support. This discovery shows that older WeChat users have the motivation to become gatekeepers of health information in their community. They want to express their experience and views about certain health issues in order to establishing their personal values or contributing to their community, which reinforces that older social media user is more likely to be a credible source of certain health issues [62].

Apart from exploring older people’ motivations for using WeChat for health-related needs, this study further examined how the motivations are associated with older users’ engagement behaviors with healthcare information on WeChat. Overall, our study confirmed that five motivations (information need, social support, surveillance seeking, social involvement and self-agency building) significantly predicted older adults’ engagement behaviors among older population. When the older people obtain health information and gain their related cognitive understanding and elaboration on it, the subsequent online behavioral responses (i.e., sharing and commenting) to health information has been triggered. Of these motives, health information needs, and social support are two important influencing factors, suggesting that the Chinese elder’s active participation in health information is driven by personal instrumental gratification [63]. This finding indicated that the gratifications of using WeChat for health purposes are high relative to the interactive activities provided by the platform (i.e., commenting on or replying to others’ health posts, sharing health articles with friends and relatives).

Technological convenience did not show any significant association with health-related engagement on WeChat. This may be because elderly simply use these embedded functions for their personal health needs such as online appointment reservation, online doctor consultation and online payment and such health needs do not require the follow-up engagement behaviors on WeChat. However, it is undeniable that WeChat offers them an easier way to fulfil people’s daily health-related needs [10].

Most notably, the sharing behavior on health information was only associated with information needs, social support, and social interaction. The other two motivations, building self-agency and surveillance, failed to predict sharing behaviors. We suspect that the older people who want to build a self-agency may tend to get more attention through liking and commenting behaviors, while those who have surveillance needs may not be keen on sharing information. This may be because they are passive participants on WeChat.

## 6. Conclusions

Chinese elderly people are increasingly turning to WeChat for health communication. However, little research has looked specifically at their motivations for health purposes, and the subsequent engagement behaviors on the site. This study, designed to fill the gap, examined the older adults’ motivations and engagement behaviors of health information on WeChat, using a sample of 690 older adults aged 50–75, and also with 20 older adults in the in-depth interview. This study examines the explanation power of U&G theory in a health context, as well as provides the practical implication for older adults’ healthcare management.

### 6.1. Theoretical and Practical Implications

This study explored the relationship between older people’s motivations and engagement behaviors, which provides the practical implication for leveraging mobile social media to improve older people’s health literacy and self-healthcare management. For instance, since building self-agency is a strong motivation for the elderly people, WeChat official account can rely more on these older opinion leaders to facilitate the dissemination of health information. Specifically, when sharing healthcare information, WeChat official account can selectively share it with some elderly people with higher influential power, and then ask them to distribute the information to their WeChat friends and groups. This may be one of the effective ways to make health information reach more target elderly people.

For theoretical implication, this study examines the explanation power of U&G theory in a health context. Most of the previous U&G research focuses on the users’ motivations and general media use and selection [64], while our research not only re-examines the explanation power of U&G in health context, but also extend the theory by providing important implications on how older people’ motives predict their subsequent engagement behaviors (i.e., sharing, liking and commenting) with health information on a social media platform. The findings indicated that U&G can be well applicable to mobile social media and health context, expanding the literature of relative U&G studies.

### 6.2. Limitations and Future Research Directions

Although we have studied the relationship between the motivations of older users and their active participation in health information on WeChat, we still do not know what kind of health information encourages them to participate. Therefore, we believe that future research will explore the types of health information to better understand older adults’ engagement behaviors. Additionally, this study ignores some factors such as health literacy and health belief. Future research should examine the influence of these factors on the behaviors of the elderly. Despite these shortcomings, this study is a positive step towards understanding the powerful role of mobile social media in promoting health communication among the older population.

## Figures and Tables

**Table 1 healthcare-09-00751-t001:** Older adult’s motivations for health purpose on WeChat.

Motivations	Frequency	Motivations	Frequency
I have acquired all kinds of medical knowledge and healthcare information through WeChat, which is difficult for me to obtain from other channels	20	To express personal stance on health issues makes me feel satisfied	10
I can obtain emotional and informational support from others on WeChat when discussing some health issues and personal health conditions	17	It is convenient and time-saving to use WeChat to book online medical registration	9
WeChat helps me to access tailored and customized healthcare advice from online disease diagnosis and treatment	15	WeChat offers online medical diagnosis and telehealth, which helps me quickly find famous doctors I would normally struggle to get an appointment with	9
I can get more health education on WeChat	15	To filter health-related misinformation and offer high quality information for WeChat friends and groups	8
WeChat helps me to access tailored and customized healthcare advice from online disease diagnosis and treatment	14	WeChat allows me to monitor and filter health-related rumors and fake news	7
I often share my experiences and knowledge on medication, daily care, diet and exercise with people with similar health conditions on WeChat	14	My community uses WeChat to issue health notifications such as vaccinations and free clinic information	7
To show my care to my family and friends by sharing meaningful health information on WeChat	13	It is very efficient and convenient for the community to deliver health notifications through WeChat	7
To build and consolidate relationships by sharing important health information	13	I feel empowered to sharing health information and to give health-related advice to WeChat friends and groups	6
Through the WeChat official account and Moments, I can keep abreast of the latest developments in public health events	12	I often ask my friends to exercise together through WeChat, which improved my enthusiasm	6
WeChat provides timely and authoritative coverage of public health issues and health policy	11	To provide professional drug information	3
I can search for any health information I need from WeChat	11	WeChat sports and health code help me manage my health	3

**Table 2 healthcare-09-00751-t002:** Demographic information of participants (*n* = 690).

Measure	Range	M (SD)	Percentage (%)
Gender	Female (vs. male)		47.5
Age in years	50–74	60.47 (6.21)	
Education	1–5 (from primary school or below to Master’s degree or above)	2.85 (0.89)	
Monthly income	1–5 (from CNY 1000 or below to CNY 7000 or above)	2.91 (1.07)	
Health status	1–5 (very bad to very good)	2.53 (0.935)	

**Table 3 healthcare-09-00751-t003:** Factor analysis of motivations for health-purpose on WeChat.

Motivation Items	Factors							
	1	2	3	4	5	6	*M*	*SD*
**Social Support**								
I can obtain emotional and informational support from others on WeChat when discussing health issues and personal health conditions	**0.765**	0.125	0.039	0.166	0.141	0.088	3.8	0.80
I often ask my friends to exercise together through WeChat, which improved my enthusiasm	**0.732**	0.221	0.104	0.14	0.136	0.135		
I often share my experiences and knowledge on medication, daily care, diet, and exercise with people with similar health conditions on WeChat	**0.731**	0.114	0.036	0.183	0.248	0.075		
I am a member of some health and wellness groups. When we talked about health-related topics, I felt the emotional support and comfort from other group members	**0.729**	0.216	0.127	0.138	0.046	0.007		
My community uses WeChat to issue health notifications such as vaccinations and free clinic information	**0.699**	0.185	0.115	0.138	0.099	0.089		
**Information Need**								
I can get more health education on WeChat	0.219	**0.788**	0.145	0.103	0.158	0.035	3.66	0.83
WeChat helps me to access tailored and customized healthcare advice from online diseases diagnosis and treatment	0.152	**0.752**	0.092	0.106	0.138	0.113		
I have acquired all kinds of medical knowledge and healthcare information through WeChat, which is difficult for me to obtain from other channels	0.197	**0.747**	0.051	0.172	0.181	0.095		
I can search for any health information I need from WeChat	0.221	**0.727**	0.103	0.128	0.158	0.12		
**Technological Convenience**								
WeChat offers online medical diagnosis and telehealth, which helps me quickly find famous doctors I would normally struggle to get an appointment with	0.12	0.062	**0.897**	0.05	0.162	0.108	3.88	0.88
It is convenient and time-saving to use WeChat to book online medical registration	0.102	0.123	**0.839**	0.063	0.187	0.154		
It is very efficient and convenient for the community to deliver health notifications through WeChat	0.101	0.155	**0.833**	0.054	0.17	0.114		
**Social Interaction**								
Through WeChat official account and Moments, I can keep abreast of the latest developments in public health events	0.314	0.15	0.052	**0.816**	0.139	0.16	3.70	0.90
WeChat allows me to monitor and filter health-related rumors and fake news	0.224	0.15	0.049	**0.812**	0.053	0.065		
WeChat provides timely and authoritative coverage of public health issues and health policy	0.136	0.151	0.068	**0.81**	0.124	0.073		
**Surveillance Need**								
To show my care to my family and friends by sharing meaningful health information on WeChat	0.179	0.191	0.188	0.093	**0.79**	0.122	3.79	0.86
To filter health-related misinformation and offer high quality information for WeChat friends and groups	0.192	0.181	0.193	0.168	**0.748**	0.117		
To build and consolidate relationships by sharing important health information	0.209	0.278	0.225	0.078	**0.72**	0.1		
**Self-Agency Building**								
I feel empowered to share health information and to give health-related advice to WeChat friends and groups	0.127	0.134	0.191	0.133	0.148	**0.891**	3.64	0.95
To express personal stance on health issues makes me feel satisfied	0.154	0.164	0.175	0.128	0.137	**0.889**		
Eigenvalue	3.19	2.733	2.487	2.254	2.084	1.783		
Variance explained (%)	15.951	13.667	12.437	11.269	10.422	8.914		
Cronbach’s alpha	0.846	0.739	0.706	0.84	0.806	0.898		

Note. Factors determined by item loading of 0.5 or higher and no cross-loadings higher than 0.5. Bold: help to clarify the results of the exploratory factor analysis.

**Table 4 healthcare-09-00751-t004:** Hierarchical regression results for motivations on engagement behaviors with health information.

	Engagement Behaviors		
	Frequency of Sharing	Frequency of Liking	Frequency of Commenting
	*β*	*β*	*β*
Block 1: Demographics			
Gender (Male = 0, Female = 1)	−0.034	−0.005	−0.007
Age	−0.155 ***	0.082 *	−0.024
Education	0.203 ***	0.182 *	0.105
Monthly Income	0.096	0.150	0.174 *
Health Status	−0.270 ***	−0.229 ***	−0.212 ***
Block 2			
WeChat usage	0.262 ***	0.274 ***	0.192 ***
Block 3: Motivations			
Social Interaction	0.179 ***	0.184 ***	0.015***
Technological Convenience	0.067	0.012	0.021
Information Need	0.251 ***	0.229 ***	0.183 ***
Social Support	0.199 ***	0.153 ***	0.155 ***
Surveillance Needs	0.085	0.136 ***	0.107
Self-agency Building	0.076	0.097 *	0.098 *
*R* ^2^	0.536 ***	0.488 ***	0.419 ***
Adjust *R*^2^	0.528 ***	0.479 ***	0.233 ***
F	65.221 ***	53.770 ***	21.882 ***

Note. *n* = 690. * *p* < 0.05; *** *p* < 0.001.

## Data Availability

Data is contained within the article or Appendix A. The data presented in this study are available in the Appendix A.

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
