# Peer review of "WeChatting for Health: What Motivates Older Adult Engagement with Health Information"

_healthcare, 2021, doi:10.3390/healthcare9060751_

Round 1

Reviewer 1 Report

In my opinion, this is an interesting study and has value in promoting the health of elderly. The analysis is done well and the conclusions support the data. I have no major comment, and only minor comments are shown below.

1. The older adults in the study were aged 50-75, but in most studies, the age threshold of the elderly sample is usually 60 or 65 years old. The author needs to explain why the age threshold of 50 is selected.

2. Many subtitles in the manuscript have duplicate numbers. For example:
    the two "2.1." on page 3 should be "2.2." and "2.3."
    "2.1." on page 4 should be "2.4."
    "2." on page 5 shoud be "3."
    "2." on page 6 shoud be "4."
    one of the two "4.2." on page 7 should be deleted
    "4.1.2." on page 9 shoud be "4.2.2."
    "2." on page 10 shoud be "5."
    "2." on page 12 shoud be "6."
    "2." on page 13 shoud be "7."
    "Table2" in the text on page 6 should be "Table 1"
     etc.

3. There are a few grammatical errors in the manuscript, such as "to further investigated" on page 1 should be "to further investigate", "to built" on page 12 should be "to build", etc.

4. There are many spelling errors in the manuscript: 
    keywords "social meida" on page 1 should be "social media"
    "conformed" on page 4 may be "confirmed"
    "inforamtion" on page 5 should be "information"
    "acount" on page 6 may be "account"
    "imformation" on page 7 should be "information"
    "social involvemet" on page 11 should be "social involvement"
    "commeting" on page 12 should be "commenting"
    improper use of "a" and "an" on page 1, 3, 5, 11
    etc.

5. There are several capitalization and punctuation errors, some of which are marked yellow in the attachment.

Minor errors similar to the above should be thoroughly and carefully checked during revision.

Author Response

Dear reviewer,thanks very much for your comments. All the problems you mentioned have been solved. Please see the attachment.

Reviewer 2 Report

Thank you for the opportunity to review your paper about WeChatting for health: What Motivates Older Adult Active Engagement of Health Information. Congratulations on the paper presented and the study described. Below is some feedback intended to help you strengthen the manuscript.

General opinion: The article is correctly written, although some information could be more summarized, so that the article is not too long. The numbering of the chapters is not well and must be corrected.

Abstract: The abstract includes the main information about the study. However, I suggest that the authors add the values of the obtained statistical results. I also suggest that the authors add the main conclusions of the study and implications of the results found.

Introduction and Background Literature: In my opinion, the Introduction and Background Literature section is well constructed and has the necessary information to understand the problem under study.

Materials and Methods: The methodology is correctly written. However, I detected some word errors that must be corrected. In relation to study 1, the authors must add the description of the ethical procedures and the description of the data analysis procedure. Regarding study 2, the authors must describe the ethical procedures and how the data analysis was carried out.

Results: Regarding study 1, the description of the data analysis found in the results section, should be moved to the methods section. I suggest the same for study 2.

Discussion: The discussion is well constructed.

Conclusions: The authors do not present the conclusions of the study. I suggest that you add this part.

References: In my opinion, the bibliography list is too long and some references are not numbered.
